# Landscape Pattern and Ecological Risk Assessment in Guangxi Based on Land Use Change

**DOI:** 10.3390/ijerph19031595

**Published:** 2022-01-30

**Authors:** Yanping Yang, Jianjun Chen, Yanping Lan, Guoqing Zhou, Haotian You, Xiaowen Han, Yu Wang, Xue Shi

**Affiliations:** 1College of Geomatics and Geoinformation, Guilin University of Technology, Guilin 541004, China; yangyp@glut.edu.cn (Y.Y.); lanyanping@glut.edu.cn (Y.L.); gzhou@glut.edu.cn (G.Z.); youht@glut.edu.cn (H.Y.); xwhan@glut.edu.cn (X.H.); wangyu@glut.edu.cn (Y.W.); shixue@glut.edu.cn (X.S.); 2Guangxi Key Laboratory of Spatial Information and Geomatics, Guilin University of Technology, Guilin 541004, China

**Keywords:** land-use change, landscape pattern, ecological risk assessment model, geographical detector, Guangxi

## Abstract

Due to ecological environmental fragility and soil erosion in Guangxi, studies of landscape patterns and associated ecological risks are needed to guide sustainable land development and ecologically sensitive land management. This study assesses dynamic spatial and temporal change patterns in land use and ecological risks based on 30 m land-use data, analyzes spatial correlations with ecological risks, and explores natural and socio-economic factor impacts on ecological risks. The results reveal: (1) A rapid and sizeable construction land increase in Guangxi from 2000 to 2018 associated mainly with loss of woodland and grassland. (2) Guangxi had the highest number of arable land patches from 2000 to 2018, and the distribution tended to be fragmented; moreover, the construction land gradually expanded outward from concentrated areas to form larger aggregates with increasing internal stability each year. (3) Guangxi ecological risk levels were low, low–medium, and medium, with significantly different spatial distributions observed for areas possessing different ecological risk levels. Regional ecological risk gradually decreased from the middle Guangxi regions to the surrounding areas and was positively correlated with spatial distribution. (4) Socio-economic factor impacts on ecological risk exceeded natural factor impacts. These results provide guidance toward achieving ecologically sensitive regional land-use management and ecological risk reduction and control, it can also provide a reference for ecological risk research in other similar regions in the world.

## 1. Introduction

In recent years, global climate change and enhanced human land development activities have led to increased risks of negative ecosystem impacts [1], such as: loss of biodiversity [2], serious environmental pollution [3], and loss of natural resources [4]. Therefore, timely prevention of potential human-induced or naturally occurring damage to ecosystems is becoming increasingly important. Ecological risk can be assessed by determining the probability of occurrence and degree of harm due to one or more natural factors or human activities that adversely affect ecosystem functions and structures [5,6]. Ecological risk assessments enable construction of whole-process, multi-level ecological risk prevention systems for application to human activities for the purpose of promoting ecosystem stability, use of virtuous cycles, and sustainable resource development [7,8]. Early ecological risk assessments predominantly addressed single risk sources and risk receptors. However, since that time such assessments have been rapidly replaced with ecological risk assessment methodologies that are no longer limited to assessing single risk sources and risk receptors, enabling scaling-up of assessment scopes from ecosystems to landscapes [9].

A landscape refers to an area of spatial heterogeneity consisting of interacting land units or ecosystems [10,11] that recur in a similar form. Landscape heterogeneity is closely related to a landscape’s capacity to recover from disturbances (resilience), as well as its stability and ecosystem diversity that together underscore the importance of the landscape as a key ecological risk assessment element [12]. Landscape ecological risk refers to possible adverse effects resulting from interactions between landscape patterns and ecological processes with natural or anthropogenic factors [13]. As compared with traditional ecological risk assessment methodologies, landscape ecological risk assessments are based on identification and analysis of coupled correlations as they relate to ecological processes and spatial patterns within landscape ecological systems. Ultimately, these assessment methodologies focus on spatial and temporal heterogeneity of risks [14] and entail comprehensive characterization and spatial visualization of multi-source risks [15].

At present, scholars have selected various assessment indicators, methods, and models for use in studies conducted in different regions with different assessment goals, all of which have yielded excellent results. In terms of assessment objects, hotspot areas for landscape ecological risk assessment research have mainly included urban areas [16], watersheds [17], administrative regions [18], nature reserves [19], etc. In terms of assessment units, landscape ecological risk assessments with different goals conducted to evaluate risks in different regions require rational selection of assessment units in order to optimize risk assessment results. There are three existing methods for dividing the assessment units: administrative districts [18], natural geographical boundaries [20], and risk cells [21]. In terms of assessment methods, landscape ecological risk assessments can be categorized into two types of methods based on risk source sinks [22] or on landscape patterns [20]. The risk source sink-based method mainly assesses risk exposures and hazards by identifying risk sources and conducting receptor analysis. By contrast, the landscape pattern method breaks away from the inherent model of traditional ecosystem assessment [23] and assesses landscape ecological risks directly from spatial patterns on a regional scale, while emphasizing integration of multi-source ecological characteristics.

Ecological risk assessment methods based on landscape patterns tend to identify and directly assess ecological risks quantitatively from the perspective of spatial landscape patterns caused by land-use changes. Land use is viewed as a comprehensive reflection of direct impacts of human economic and social activities on surface resources and the natural environment [24], whereby spatio-temporal heterogeneity of land use is influenced by regional topographic and geomorphic features that are closely tied to spatio-temporal changes in landscape patterns and landscape ecological risks [25]. Currently, development of ecological risk assessment models based on land-use changes is a research hotspot [26]. For example, Xie et al. [27] analyzed ecological risks associated with landscape patterns within the region of Poyang Lake from 2010 to 2018 and ranked landscape pattern risks in descending order as watershed, construction land, unused land, woodland, grassland, and arable land. As another example, Zhang et al. [28] analyzed land-use changes and dynamic characteristics of coastal cities in China using a land-use transfer matrix and other tools so as to further analyze the impact of land-use changes on landscape ecological risks.

Guangxi possesses a relatively fragile natural ecological environment characterized by high mountains and steep slopes, poor soil, severe soil erosion, and frequent natural disasters [29]. In order to reduce environmental risks, Guangxi has been vigorously implementing seven key national ecological projects, including afforestation, pollution prevention, and control and comprehensive management of stone desertification. In addition, efforts have been made to continuously improve natural ecosystem balance through the implementation of a series of institutional reforms and policies. Nevertheless, rapid industrialization and urbanization in that region [30] have seriously damaged arable land, woodland, and water bodies and continue to undermine attainment of ecological security in Guangxi [31]. Therefore, it is particularly important that ecological risk assessments are conducted in Guangxi in order to guide efforts toward establishing a systematic and effective ecological environment monitoring and early warning system.

This study constructs an ecological risk assessment model according to both the landscape disturbance index and landscape fragility index, while also taking into account land-use changes toward the goal of comprehensively describing the overall spatial pattern characteristics of ecological risk in Guangxi. At the same time, in order to monitor land-use changes as they occur with time, a geographic detector is used to quantitatively analyze the driving causes of landscape ecological risk changes from both natural and socio-economic perspectives. Moreover, in order to guide ecologically sensitive land development in Guangxi, magnitudes of driving forces for each ecological risk factor and results of ecological risk autocorrelation analysis are taken into account.

## 2. Materials and Methods

### 2.1. Study Area

Guangxi is located next to the southwestern border of China (104°26′–112°04′ E, 20°54′–26°24′ N) and encompasses a total area of 236,700 km^2^ [32]. Guangxi abuts the southeastern edge of the Yungui Plateau and is bordered by Guangdong to the east, the Gulf of Tonkin to the south, and the sea surrounding Hainan and Yunnan to the west (Figure 1). Topographically, land elevation in Guangxi, which is high in the northwest and low in the southeast, follows a northwest to southeast slope with a basin-like depression in the middle region that is surrounded by mountains and few plains (Guangxi Basin) [33]. Guangxi has a unique karst landscape with widely distributed stone desertification areas that are concentrated in the southwest, northwest, central, and northeast areas of the region and that together account for about 37.8% of the total area of Guangxi. Due to its location at low latitudes, Guangxi has a central subtropical monsoon climate and southern subtropical monsoon climate with an average annual temperature above 16 °C, average annual rainfall above 1100 mm, with high temperatures and precipitation levels in summer and short sunshine hours and dry and warm weather in winter [34]. Due to its unique geographical location, Guangxi is located within the South China Economic Circle, Southwest China Economic Circle and ASEAN Economic Circle and is an important gateway node of China’s “One Belt, One Road” global infrastructure development strategy [35].

### 2.2. Data Source

Land-use data with spatial resolution of 30 m were used in this study and were obtained from the Resource and Environment Science and Data Center of the Chinese Academy of Sciences (https://www.resdc.cn/ (accessed on 14 January 2021)). Six primary land-use types (arable land, woodland, grassland, water, construction land, and unused land) and 25 secondary land types are studied in this work. In order to rigorously assess temporal change characteristics of landscape and ecological risks in Guangxi, here we use data of land use in Guangxi for the years 2000, 2010, and 2018. ArcGIS is used to map the six aforementioned primary land-use types for use in analysis (Figure 2).

Meteorological data with a spatial resolution of 0.5° were obtained from the European Centre for Medium-Range Weather Forecasts (ECMWF) as third-generation reanalysis information accessed through the ERA-Interim website (https://apps.ecmwf.int/datasets/data/interim-full-daily/levtype=sfc/ (accessed on 16 December 2020)). Digital elevation model (DEM) data with 30 m spatial resolution are derived from geospatial data cloud (http://www.gscloud.cn/ (accessed on 7 March 2021)). Although numerous types of topographic factors may impact ecological risk, based on the topography of Guangxi and results of previous studies, elevation and slope factors appear to considerably influence ecological risk in the region [36], prompting us to explore the effects of these factors on ecological risk in this work. We also incorporate additional data in our analysis, such as the leaf area index, gross domestic product (GDP), and population density data obtained from the Resource and Environment Science and Data Center of Chinese Academy of Sciences. The spatial resolution of the leaf area index data was 8 km, while spatial resolutions of GDP and population density data were each 1 km. For all of the abovementioned data sources, 2000, 2010, and 2018 data were used, with exceptions for GDP and population density data, for which 2015 data were selected instead of 2018 data due to limitations and little fluctuations of 2018 data. For further analysis, the above data were resampled to 30 m and the projection coordinate system was unified as Krasovsky_1940_Albers.

### 2.3. Research Methodology

This study explores the spatial and temporal changes in land use, landscape patterns, and ecological risks in Guangxi, specifically analyzing the formation, changes, and linkages of ecological risks from multiple perspectives. At the same time, it provides reference suggestions for ecological construction in Guangxi based on the research results, avoiding the phenomenon whereby previous studies have focused on assessment rather than on application. The overall methodology of this study is shown in the flow chart (Figure 3), mainly divided into the following three points: (1) analyzing the spatial and temporal characteristics of land-use change in Guangxi based on land-use change degree and land-use transfer matrix; (2) studying the ecological risk change in Guangxi using the comprehensive ecological risk index model; (3) exploring the spatial correlation of ecological risk in Guangxi using the autocorrelation analysis method, and at the same time, exploring the impact of each driver on ecological risk using the geographical detector method.

#### 2.3.1. Land-Use Change

(1) Land-use change metrics

Land-use dynamics is an indicator used to describe quantitative area changes in land-use types in a region over a certain period of time to reveal the magnitude of regional land-use change and compare differences between different landscape types during different time periods or between different regions [37]. In this study, the single land-use dynamic attitude of each land-use type in Guangxi during different time periods from 2000 to 2018 was calculated to analyze the land-use changes in the study area. The formula used to calculate dynamic attitude is shown below:(1)$$K=\frac{U_{b}-U_{a}}{U_{a}}\times \frac{1}{T}\times 100\%$$
where *K* is the dynamic attitude of a specific land-use type during the study period; *U_a_* and *U_b_* are areas of a specific land type at the beginning and end of the study period, respectively; *T* is the length of the study period in years, with *T* = 19 in this study.

(2) Land-use transfer matrix

The land-use transfer matrix can be used to monitor the interconversion between land-use types in the study area within a fixed time period, indicating the direction of transfer and the area of conversion of each land-use type, which can further reveal spatial characteristics and evolutionary patterns and associated mechanisms that drive land-use changes [38]. The land-use transfer matrix formula is shown below:(2)$$S_{ij}=\left[\begin{array}{cccc} S_{11} & S_{12} & \cdots & S_{1n}\\ S_{21} & S_{22} & \cdots & S_{2n}\\ \cdots & \cdots & \cdots & \cdots \\ S_{n1} & S_{n2} & \cdots & S_{nn} \end{array}\right]$$
where, *S_ij_* is the number of type *i* land use at the beginning of the study and of type *j* land use at the end of the study, where *i* and *j* are land-use types in Guangxi at the beginning and end of the study period, respectively; *n* is the total number of land-use types, with *n* = 6 in this study.

#### 2.3.2. Ecological Risk Assessment Model

Previous studies have shown that when the size of the sample square reaches 2 to 5 times the average area of landscape patches in the study area, calculation of sample square area can fully reflect comprehensive landscape pattern information around the sampling points [39]. Therefore, to fully demonstrate the spatial divergence of landscape indices and ecological risks in Guangxi, in this study, based on the actual situation, we divided Guangxi into 682 ecological risk assessment cells of area 20 km × 20 km that comprised a square-based grid. This grid was used to perform equal spacing sampling then the ecological risk index was calculated for each cell in the grid using Fragstats 4.1 software to generate the landscape grid-based ecological risk index model. Finally, ArcGIS was used with the ordinary kriging interpolation method to draw the spatial distribution map of ecological risk in Guangxi.

The landscape pattern index and ecological risk have some correlation and connectivity; therefore, in order to establish the relationship between landscape structure and ecological risk, the landscape disturbance index (*E_i_*) and landscape vulnerability index (*F_i_*) were selected in this study for constructing a comprehensive Ecological Risk Index (*ERI*) model for Guangxi [15]. The landscape ecological risk index was calculated as follows [40]:(3)$$ERI_{k}=\sum_{i}^{n}\frac{S_{ki}}{S_{k}}\sqrt{R_{i}}$$
where *ERI_k_* is the ecological risk index of the *k*-th risk plot; *S_ki_* is the area of the *i*-th landscape type of the *k*-th risk plot, *S_k_* is the total area of the *k*-th risk plot, and *R_i_* is the loss degree index of the *i*-th landscape type. Additional specific formulas and descriptions are shown in Table 1:

The natural breakpoint method was used in ArcGIS to classify the landscape ecological risk of the study area into five levels: low, medium–low, medium, medium–high, and high risk.

#### 2.3.3. Spatial Analysis Method

Spatial autocorrelation analysis is a statistical method used to detect the degree of correlation between variables in an assessment unit and its neighboring unit variables [46]. Here, spatial autocorrelation analysis includes global spatial autocorrelation and local spatial autocorrelation, which are represented by Moran’s I index (*I*) and LISE index (*I_i_*), respectively [45]. Moran’s I index is a widely used spatial autocorrelation statistic that can reflect the overall spatial association and difference status of landscape ecological risk in Guangxi, and higher spatial autocorrelation is a prerequisite for spatial interpolation of landscape ecological risk. In this study, GeoDa software was used to analyze global and local spatial autocorrelation statistics associated with ecological risk in Guangxi using the following formula:(4)$$I=\frac{n\sum_{i}\sum_{j}\omega_{ij}\left(Y_{i}-\bar{Y}\right)\left(Y_{j}-\bar{Y}\right)}{\left(\sum_{i\neq j}\omega_{ij}\right)\sum_{i}{\left(Y_{i}-\bar{Y}\right)}^{2}}$$
where *Y_i_* and *Y_j_* are values of variables in adjacent paired cells; *ω_ij_* is the spatial weight matrix; *Y* is the mean of attribute values; *i* takes values between (−1,1). When *I* > 0, this indicates that the observed objects in the study cell tend to be spatially aggregated and positively spatially correlated; when *I* < 0, this indicates a discrete spatial distribution that is negatively spatially correlated; when *I* = 0, this indicates no spatial correlation.

The LISA index, also known as the local Moran’s I index, reflects the degree of difference and significance between a region and its neighboring regions and is calculated using the following formula:(5)$$I_{i}=\frac{Y_{i}-\bar{Y}}{S^{2}}\sum_{j\neq i}^{n}\omega_{ij}\left(Y_{j}-\bar{Y}\right)$$
where *n*′ is the sample size expressed as the number of study units, and *S*_2_ is the variance of the statistic. When *I_i_* > 0, this means that a region with high (low) observations is surrounded by a region with high (low) observations, which is equivalent to “high-high” (“low-low”) aggregation; when *I_i_* < 0, this means that a region with high (low) observations is surrounded by a region with low (high) observations, which is equivalent to “high-low” (“low-high”) aggregation; when *I_i_* = 0, this means that the observed region is not associated with the neighboring region, which is equivalent to not significant.

#### 2.3.4. Geographical Detector Method

The geographical detector method is a set of statistical methods that detect spatial differentiation, while also revealing driving forces behind it [47]. The factor detection method can be used to analyze the magnitude of the explanatory power of each driver on the spatial and temporal variations in ecological risk of the landscape [48] and is calculated using the following formula:(6)$$q=1-\frac{\sum_{h=1}^{L}N_{h}\sigma_{h}^{2}}{N\sigma^{2}}$$
where *q* is the strength by which a factor explains spatial and temporal variations in ecological risk of the landscape, with *q* taking values between (0,1); *q* = 0 indicates that ecological risk is randomly distributed such that the larger the *q* value of a factor, the stronger the explanatory power of the factor on ecological risk in the landscape; *N* is the number of samples in the study area; *σ*^2^ is the variance of the index; *h* is the index grading, and *L* is the number of graded layers (e.g., *h* = 1, 2, and… *L*).

In this study, natural factors (temperature, precipitation, elevation, slope, and leaf area index) and social factors (GDP, population density) were selected to quantitatively determine the magnitude of the contribution of each driver to the change in landscape ecological risk in Guangxi. Among them, natural factors are discretized using the natural breakpoint method and social factors are discretized using the equivalence method, with values for each factor assigned to 9 levels.

## 3. Results and Analysis

### 3.1. Land-Use Change and Landscape Characteristics

#### 3.1.1. Land-Use Types Change

The total areas of each land-use type in Guangxi are ranked in descending order as woodland > arable land > grassland > construction land > water area > unused land (Table 2). From 2000 to 2018, the area of each land-use type in Guangxi changed according to different trends, with the area of arable land showed a decreasing trend following an increasing rate of decrease, areas of water, and construction land showing increasing trends, the area of woodland showing a small increase and then a decrease and grassland and unused land areas showing a decreasing and then increasing trend. Areas of construction land, water, and unused land increased by 1652.79 km^2^, 214.31 km^2^, and 50.99 km^2^, respectively, while areas of arable land, woodland, and grassland decreased by 1006.49 km^2^, 534.28 km^2^, and 318.08 km^2^, respectively.

From 2000 to 2010, a total of 2932.65 km^2^ of land in Guangxi was transformed into other land-use types (Table 3), with transformation of arable land occurring to the greatest extent, reaching 1061.33 km^2^ that accounted for 36.19% of the total transformed land area; this change was mainly due to the conversion of arable land into woodland and developed land. Transformation of woodland to other land types ranked next in extent, reaching 922.54 km^2^ that accounted for 31.46% of the total transformed area. Overall, areas of woodland and arable land increased by 1033.26 km^2^ and 806.23 km^2^, respectively. During 2000–2010, construction land area increased significantly (by 343.61 km^2^) that was mainly due to conversion of woodland and grassland to construction land.

A total of 5452.42 km^2^ of land in Guangxi underwent land-use type shifts from 2010 to 2018 (Table 4), with more pronounced fluctuations in land-use type observed as compared to changes that occurred from 2000 to 2010. Losses of arable land and woodland were greatest, amounting to 2149.47 km^2^ and 2119.24 km^2^, respectively, which accounted for 39.42% and 28.87% of the total area affected by land-use shifts, respectively. Arable land was mainly converted into woodland and construction land, while woodland was mainly converted into arable land and grassland. The largest shift in land-use area was due to an increase of 1509.65 km^2^ in construction land that accounted for 27.69% of the total transformed land area, followed in increasing order by area by woodland, arable land, and grassland increases. During 2010–2018, arable land and woodland areas decreased significantly, with losses reaching 751.36 km^2^ and 646.71 km^2^, respectively, while other land-use types showed an increasing trend, with construction land area increasing significantly by 1306.99 km^2^.

#### 3.1.2. Landscape Pattern

The landscape pattern of Guangxi changed significantly from 2000 to 2018 (Table 5), with the number of patches continually increasing each year from 136,854 in 2000 to 139,459 in 2018 (except for a decrease in number of water patches). Numbers of patches for other land-use types showed increasing trends, among which the number of developed land patches increased significantly. The number of arable patches was highest; although, the total arable land area decreased significantly as fragmentation and separation indices increased and the dominance index decreased. The woodland area was widely distributed and had the largest dominance index and significantly smaller fragmentation and separation indices as compared to other land-use types. Meanwhile, the area of grassland decreased and the number of grassland patches increased, as did fragmentation and separateness indices, while opposite trends were observed for water areas. Overall, fragmentation index and separation index trends for each land-use type were consistent and opposite to dominance index trends.

### 3.2. Analysis of Spatial and Temporal Changes in Ecological Risks

#### 3.2.1. Spatial and Temporal Evolution of Ecological Risks

According to the experimental results, the ecological risk value of Guangxi is between 9.74 and 12.80, and the ecological risk grade assignment interval is: low risk (<10.22), low–medium risk (10.22~10.57), medium risk (10.57~10.98), medium–high risk (10.98~11.16), and high risk (>11.16). Ecological risk of Guangxi in 2000–2018 was dominated by low risk, low–medium risk, and medium risk areas (Figure 4), which occupied 28.29%, 26.14%, and 23.91% of the total area of Guangxi, respectively; areas with medium–high and high levels of risk occupied smaller proportions that were 13.67% and 7.99%, respectively, of the total area of Guangxi. Change trends for different ecological risk levels in Guangxi from 2000 to 2018 differed, with areas with low, medium–high, and high levels of risk showing increasing trends and medium- and low–medium-risk areas showing decreasing trends. From 2000 to 2018, areas with medium and medium–high levels of risk changed by 10,817.95 km^2^ and 7927.81 km^2^, respectively.

The overall ecological risk level in Guangxi from 2000 to 2018 was not high, but the spatial distribution of risk varied significantly (Figure 5). Moreover, ecological risk levels of Guangxi followed a trend of gradual decrease from the middle of the region to the edges. High-risk areas were mainly concentrated in Nanning, Guigang, and Laibin in central Guangxi, and Beihai and Qinzhou in southern Guangxi; medium–high-risk areas were mainly concentrated in Nanning, Qinzhou, and Liuzhou areas; low-risk areas were mainly concentrated in areas of Baise, Hechi, and other areas. During 2000–2018, high-risk and medium–high-risk areas showed clear trends of expansion, where the spatial distribution of medium–high-risk areas reflected gradual coalescence of areas to form interconnected patches.

#### 3.2.2. Ecological Risk Land Class Distribution

Construction land was mainly distributed within the medium-risk area in 2000 (Figure 6), accounting for 35% of the total construction land area, while in 2010 and 2018 construction land was mainly distributed in the medium–high-risk area, accounting for 30% and 31%, respectively. In 2000, unused land was mainly distributed in lower risk areas that accounted for 50% of the total unused land area, while in 2010, unused land was evenly distributed in areas of low–medium, medium–high, and high risk. In 2018, unused land was mainly distributed in high-risk areas that accounted for 77% of the total unused land area, with unused land present mainly as separate patches with unstable internal structure and fluctuating ecological risk. In 2000, water areas were mainly distributed in medium-risk areas that accounted for 28% of the total water area. By contrast, in 2010 and 2018, water areas were mainly distributed in areas of high risk that accounted for 26% and 25% of water area, respectively, indicating that the ecological risk level of the area where water bodies are located had increased. Nevertheless, water areas do not occupy a large proportion of areas with high risk overall and their ecological risk level distribution tends to be uniform. From 2000 to 2018, grassland and arable land areas were mainly distributed in areas with medium risk, while woodland was mainly distributed in areas with low risk; low proportions of grassland and woodland areas were found in high-risk areas.

#### 3.2.3. Autocorrelation Analysis

The Moran’s I of ecological risks in Guangxi from 2000 to 2018 was greater than zero (Figure 7), showing a positive spatial correlation, while ecological risk interactions with each other exhibited spatial similarity. Moran’s I values were 0.431, 0.484, and 0.305 in 2000, 2010, and 2018, respectively, with values showing an upward and then downward trend and an overall decrease during the 2000–2018 period.

Spatial aggregation patterns of ecological risk values in Guangxi can be divided into five categories, high-high aggregation (H-H), low-low aggregation (L-L), high-low aggregation (H-L), low-high aggregation (L-H), and non-significant aggregation (N-S) (Figure 8). For 2000, 2010, and 2018, proportions of high-high spatial aggregation patterns were 14.24%, 15.84%, and 13.81%, respectively, while corresponding proportions of L-L aggregates were 23.40%, 24.71%, and 22.67%, respectively. Both H-H aggregation areas and L-L aggregation areas followed a trend of an initial increase followed by a decrease with an overall decreasing trend. In terms of spatial distribution, H-H aggregation areas were mainly concentrated in central and southern areas of Guangxi, where their spatial distributions coincided with distributions of areas with medium–high and high ecological risk. By contrast, L-L aggregation areas were scattered throughout marginal areas of Guangxi, where their spatial distributions coincided with areas with low ecological risk.

#### 3.2.4. Landscape Ecological Risk Impact Factor Analysis

The explanatory power of each factor influencing ecological risk in Guangxi landscapes from 2000 to 2018 was analyzed using a geographical detector method. Explanatory power values of factors in 2000 were ranked in descending order as GDP (0.2073) > population density (0.2049) > leaf area index (0.2027) > elevation (0.1880) > temperature (0.1747) > slope (0.1393) > precipitation (0.1087), with GDP explanatory power greater than that of the other factors (Figure 9). Elevation explanatory power increased from 0.1880 in 2000 to 0.2316 in 2010, with this large increase indicating that elevation had become one of the leading causal factors of landscape risk in Guangxi by 2010. By contrast, precipitation and leaf area indices showed small increases in explanatory power as compared to all other influencing factors. In 2018, GDP, elevation, and precipitation explanatory power values were >0.2, while slope, leaf area index, and population density values all showed small decreases relative to their corresponding values in 2010 and temperature explanatory power increased from 0.1858 in 2010 to 0.2091 in 2018. Overall, socioeconomic factors had a greater impact on ecological risk than natural factors.

## 4. Discussion

### 4.1. Analysis of Land-Use Change and Landscape Characteristics

Due to the combined influence of natural and human factors, land use in each region of Guangxi has been dynamically changing [49]. According to third-generation reanalysis data provided by the ERA-Interim website of the European Centre for Medium-Range Weather Forecasting, the annual average temperature of Guangxi increased by 0.5 °C and the annual average amount of precipitation increased by 259.16 mm from 2000 to 2018, with such changes expected to drive many natural factor changes. In addition, according to the Guangxi Statistical Yearbook, it is known that the total population of Guangxi in 2000 was 47.51 million and the per capita gross regional product was RMB 4652 as compared to the total population in 2018 of 56.59 million and the per capita gross regional product of RMB 40,012. Such increases in social and economic activities are also expected to impact natural factors, since drivers of land-use change are closely related to natural environmental changes and human activities. Moreover, these results demonstrate how land-use change studies can be used to visually reveal environmental impacts of human activities [50].

Different land-use types have different forms and rates of change due to their different uses and functions [51]. The large area of steep-slope arable land in Guangxi and the existence of a large amount of rock-desert arable land and arable land undergoing soil erosion [52] have led to an increased ecological risk of arable land in the region. In addition, due to effects of economic and social development and occurrences of natural disasters, arable land is being gradually replaced with construction land, resulting in a significantly decreased regional arable land area, an increase in arable land patch number and increased arable land fragmentation. Meanwhile, woodland has various ecological functions, such as water conservation [53], soil and water conservation [54], and species protection [55], etc. Guangxi woodland covers a large area that is mostly concentrated in patches, with significant dominance trends and small change trend that keep ecological risk low–medium. Nevertheless, in the 21st century, peak periods of industrialization, urbanization, and infrastructure development have occurred in Guangxi that have increased the demand for land development, which has remained high for quite a long time [56]. In fact, construction land area increased by 1306.99 km^2^ from 2000 to 2018, due to development of large amounts of original woodland and grassland areas. Moreover, construction land area is growing significantly faster than the number of patches, due to rapid development of the area that has resulted in the formation of patches in a concentrated and contiguous manner that has led to enhanced patch aggregation and increased internal stability. Meanwhile, water areas serve an important ecological function by maintaining the balance and stability of ecosystems. There are numerous rivers in Guangxi that belong to four major watersheds, namely, Pearl River, Yangtze River, Baedu River, and Binhai that have increased in total area due to effective management at the local level.

Different types of land use in each region determine different forms of local economic development and ecological protection [57,58]. Guangxi is mainly dominated by forest land, arable land, and grassland. Forest land and grassland are mainly distributed in patches in Baise and Hechi in northwestern Guangxi and Hezhou, Wuzhou, and Yulin in eastern Guangxi. These areas are mainly dominated by forestry and pastoralism such that forest land and grassland play important roles in the economic development and ecological stability of the region. Northwestern Guangxi is rich in mineral resources, but the region is relatively economically poor and has serious rock desertification problems that should be controlled; thus, in this region active development of unused land should be promoted to provide land for urban construction. Arable land and construction land are mainly distributed in Nanning, Guigang, and Laibin in central Guangxi, and in Beihai and Qinzhou in southern Guangxi, which are mainly agricultural and industrial areas with leading roles in the economic development of Guangxi. These areas are also are rich in water resources and have supported moderate land development, but these areas should be improved and maintained as arable land.

### 4.2. Ecological Risk Change Analysis

Ecological risk levels of areas of Guangxi in 2000–2018 mainly included low, low–medium, and medium risk levels, with overall increases in risk levels observed. However, spatial distributions vary significantly for each risk level, with risk levels that are high in the south, low in the north, high in the middle, and low in all directions outward from the middle. Areas with high ecological risk areas are concentrated in Laibin, Guigang, and Nanning in central Guangxi, Qinzhou, Beihai, and Fangchenggang and other areas in southern Guangxi, while areas with low ecological risk are mainly concentrated in northern Baise, Hechi, and other areas in the north. Importantly, the spatial distribution of ecological risks is closely related to regional natural conditions and socio-economic conditions [59], with different natural conditions determining which industries are suitable for regional development that, in turn, will shape land use and human activities in that region [60]. The southern and central regions of Guangxi have medium–high temperatures, low levels of precipitation, low elevation and gradual slopes, high population, high GDP, and high distributions of construction land and unused land. These factors contribute to the medium–high ecological risk in Guangxi, with the southern region possessing especially high ecological risk due to its status as a coastal region. As compared to 2000, areas with high and medium–high risk within Nanning and Qinzhou and other places underwent rapid expansion by 2010 that may have due to economic developmental pressures that led to a continuous increase in construction land with associated high ecological risk. It is worth noting that land areas with high ecological risk in Nanning in 2010 decreased in size significantly by 2018, with reductions in land areas concentrated in Jiangnan District and Xixiangtang District of Nanning. These changes are due to good progress made in those regions since 2010 in implementing ecological initiatives, such as “ecological village” construction, afforestation, water environment management, and village greening.

In order to scientifically maintain the ecological security of Guangxi, Guangxi is divided into ecological risk key control zones, strict control zones, and general control zones in which high-risk areas and medium–high-risk areas correspond to key control zones, while areas with medium and low–medium risk correspond to strict control zones and low-risk areas correspond to general control zones (Table 6).

Ecological risk key control zone: This area is mainly concentrated in the central and southern part of Guangxi, with low elevations and gradual slopes, low levels of precipitation and high temperatures. Due to rapid economic development and GDP growth, these areas contain high proportions of construction land that are rapidly increasing, resulting in increased ecological pressure on the local ecological environment that causes ecological risk to continually rise. Therefore, in these areas, it is necessary to develop reasonable ecological red line policies, raise awareness of local governments and residents regarding the importance of ecological risk mitigation, strictly control the scale of urban land development, promote transformation of land-use types, strengthen water pollution control, and enhance protection of ecological systems such as forests, grasslands, and high-quality arable land.

Strictly controlled ecological risk zones: Distribution of various land-use types in this zone are balanced and impacts of natural and socio-economic factors are low as compared to their impacts in other zones. Therefore, these zones should play a leading role in land-use planning by coordinating current land-use distribution patterns, strengthening optimization and integration of construction land and by rationally developing land according to resource availability to promote economic development and reduce ecological risks at the same time.

General control ecological risk zones: These areas are mainly scattered along the margins of Guangxi and are characterized by high mountains and steep slopes, high levels of precipitation, low temperatures, a relatively low economic level, predominance of woodland and grassland, relatively little human activity, and low ecological risk. Therefore, the effective arable land area should be increased, the quality of arable land should be improved, construction land should be developed moderately and reasonably, and infrastructure development should be strengthened. In addition, advantages of local woodland distribution should be maintained, while comprehensive management of stone desertification and karst areas should be vigorously carried out to repair and improve the ecological health of the environment.

### 4.3. Study Shortcomings and Recommend Processed Improvements

First, the validity of the evaluation used in this study is uncertain, due to differences in data quality [61], risk cell selection [20], and weight assignments [62] that all increased uncertainty. Accuracy and quality of remote sensing images and the accuracy of interpretation largely affect data quality, with different criteria used to define risk cells producing different scale effects and variable evaluation results due to use of different weight assignment methods. This study adopted the equally spaced grid method to divide risk cells, a method that is conducive to spatial interpolation, which is necessary to generate a risk spatial distribution map. However, this method disrupts the continuity of the original natural landscape. Secondly, when constructing the comprehensive ecological risk index model of Guangxi, assessment results obtained using different assignment methods will differ. Therefore, future research is needed to develop more powerful uncertainty analysis methods that focus on analyzing the uncertainty of each link in the assessment process for use in improving the reliability of assessment results.

Second, evaluation indicators and evaluation criteria are not uniform and therefore cannot serve as unified evaluation indicators for use in an ecological risk assessment, as selection of evaluation indicators that vary significantly would lead to different selection of assessment objects. At the same time, although the classification of ecological risk levels is generally not dependent on research needs, distribution ranges of ecological risk values and classification intervals differ greatly among different research disciplines. Therefore, ecological risk assessment results are relative and can only be used to assess relatively high and low regional ecological risks.

Finally, ecological risk management research efforts have been relatively weak [63], due to the lack of a mature ecological risk management framework system. Meanwhile, research to date has mainly focused on risk assessment itself and not on the formulation of implementable recommendations based on assessment studies geared toward mitigating regional ecological risk.

## 5. Conclusions

(1) Guangxi land-use types mainly include woodland, arable land, and grassland. Woodland and grassland are distributed in patches within marginal areas of Guangxi, while arable land and construction land types are mainly concentrated within the middle zone. Due to influences of natural factors and human activities, large areas of arable land, woodland, and grassland have been developed, such that each patch of construction land tends to be located within larger areas of concentrated land patches that tend to rapidly increase in area with time.

(2) The overall ecological risk level of Guangxi is low, but differences in spatial distribution of ecological risks are significant; the northwest half of Guangxi shows a trend of gradually increasing ecological risk when analyzing land areas from the edges toward the central area, while the southern half that is adjacent to the South China Sea shows steadily increasing ecological risk. In terms of spatial distribution, areas of high risk and medium–high risk gradually expand to form a continuous distribution. Combining land-use types, woodland is mainly distributed in low-risk areas, while arable land and grassland are mainly distributed in areas with medium ecological risk, and construction land, water, and unused land are distributed in areas with highest ecological risk.

(3) Ecological risks in Guangxi from 2000 to 2018 show positive correlations, with Moran’s I index values rising then falling and the spatial aggregation and spatial differentiation of ecological risks diminishing overall. In terms of spatial distribution, high-high aggregation is mainly concentrated in the central and southern coastal areas and tends to be focally concentrated, while low-low aggregation is distributed sporadically throughout peripheral areas.

(4) From 2000 to 2018, influences of socio-economic factors on ecological risk were greater than influences of natural factors, with GDP and population density acting as important drivers of changes in spatial and temporal distributions of ecological risk across the landscape of Guangxi.

## Figures and Tables

**Figure 1 ijerph-19-01595-f001:**
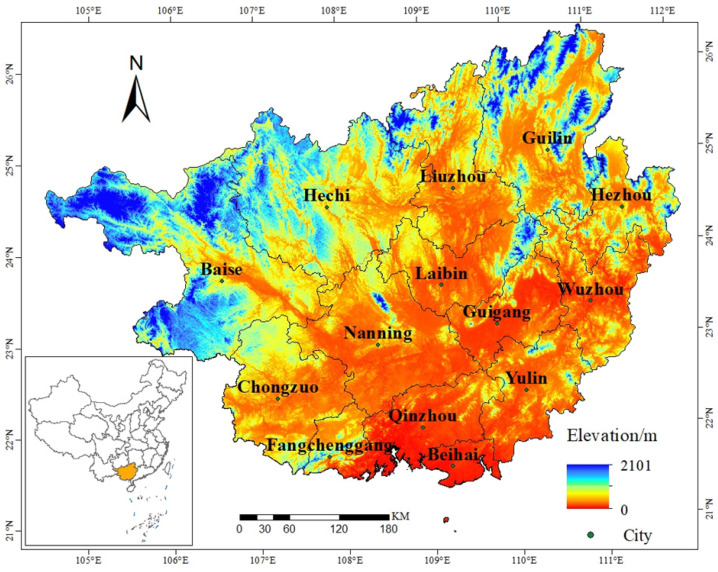
Location of the study area.

**Figure 2 ijerph-19-01595-f002:**
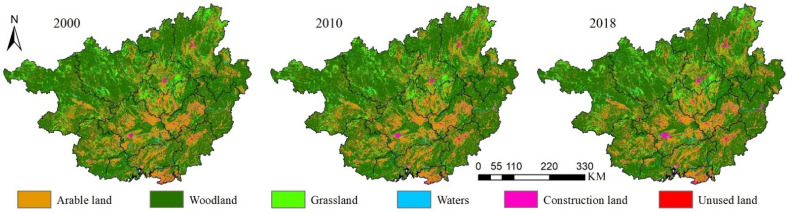
Land-use types in Guangxi.

**Figure 3 ijerph-19-01595-f003:**
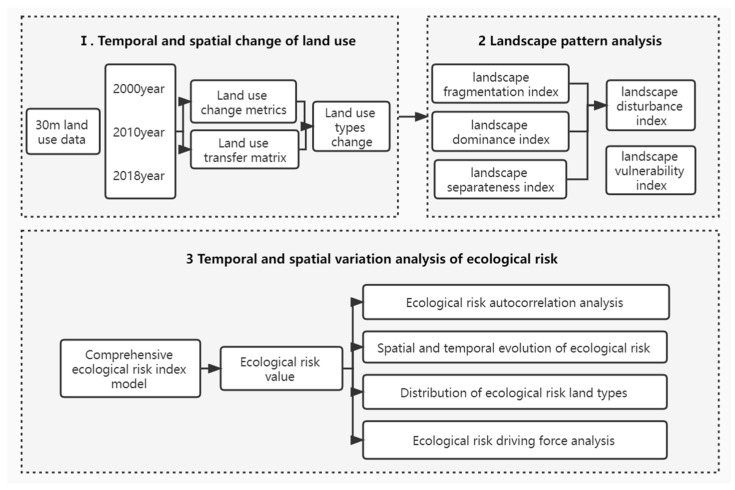
The framework of the research.

**Figure 4 ijerph-19-01595-f004:**
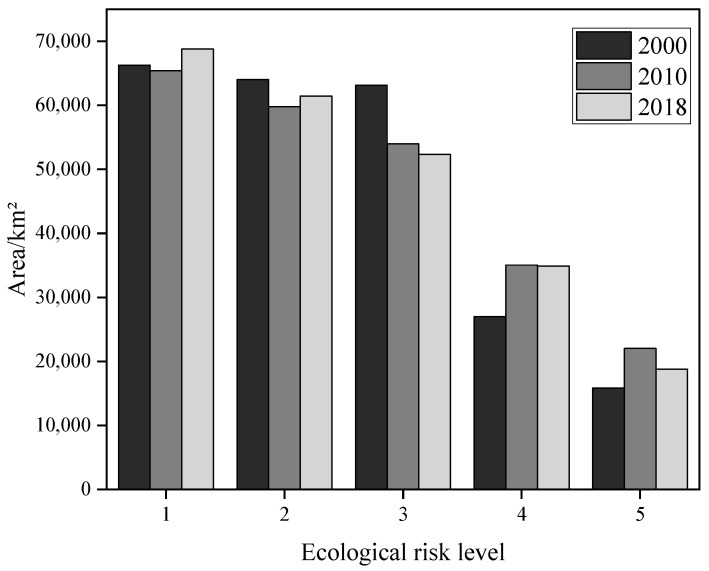
Area map of ecological risk level in Guangxi. Note: 1, 2, 3, 4, and 5 represent low risk, low–medium risk, medium risk, medium–high risk, and high risk, respectively.

**Figure 5 ijerph-19-01595-f005:**
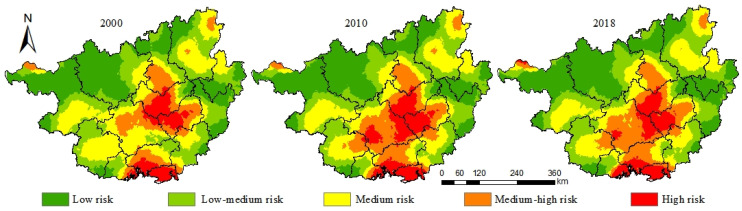
Spatial distribution of ecological risk levels in Guangxi.

**Figure 6 ijerph-19-01595-f006:**
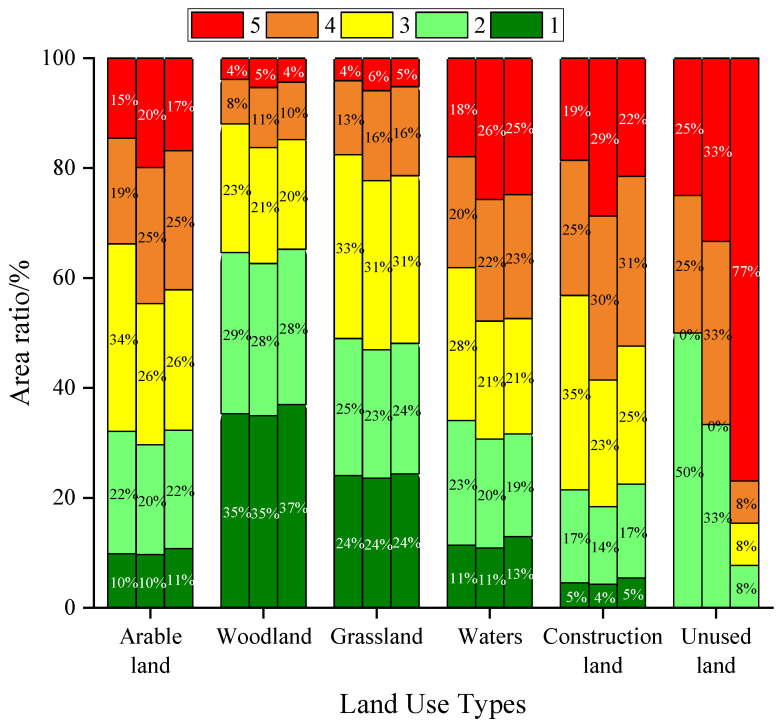
Distribution of ecological risks by land-use types. Note: 1, 2, 3, 4, and 5 represent low risk, low–medium risk, medium risk, medium–high risk, and high risk, respectively.

**Figure 7 ijerph-19-01595-f007:**
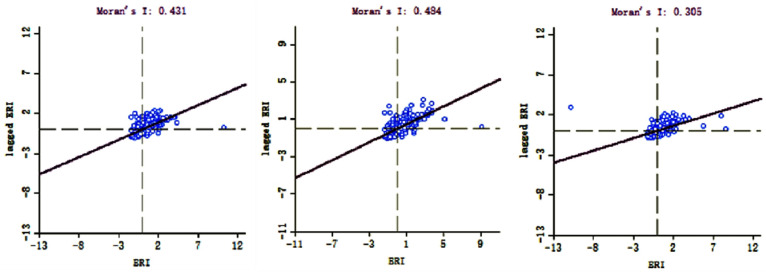
Global spatial autocorrelation of ecological risk in Guangxi.

**Figure 8 ijerph-19-01595-f008:**
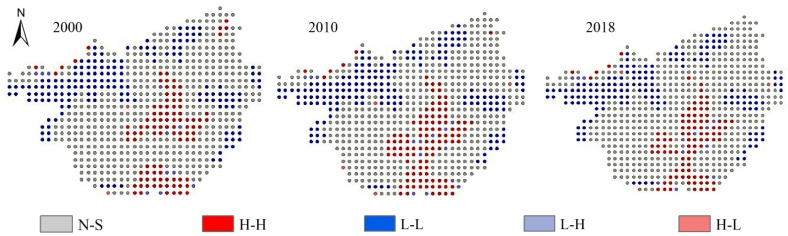
Local autocorrelation diagram of ecological risk.

**Figure 9 ijerph-19-01595-f009:**
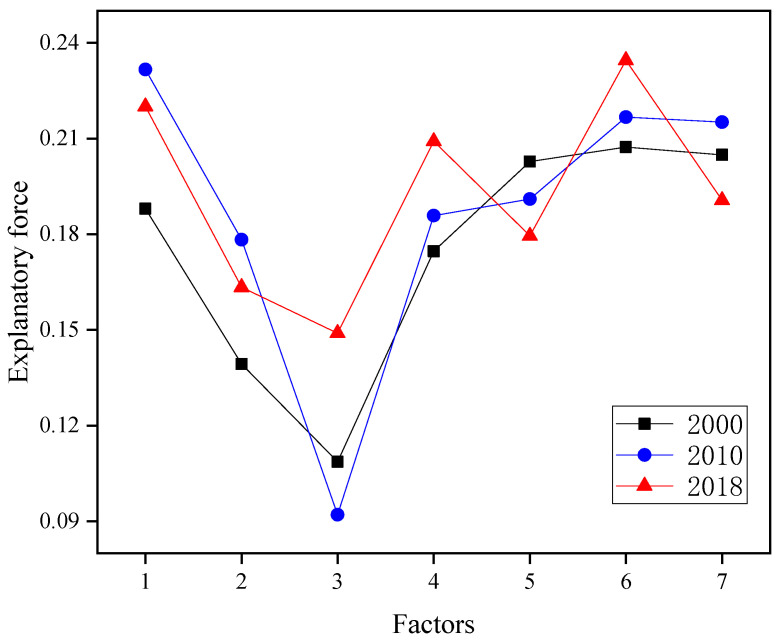
Map of landscape ecological risk influencing factors. Note: 1, 2, 3, 4, 5, 6, and 7 denote elevation, slope, precipitation, temperature, leaf area index, GDP, and population density, respectively.

**Table 1 ijerph-19-01595-t001:** Calculation formula of landscape index and ecological significance.

Index	Calculation Formula	Ecological Significance
Landscape fragmentation index (*C_i_*)	$C_{i}=\frac{n_{i}}{A_{i}}$	Describes the degree of fragmentation of a landscape type in the region at a given time; such that, the higher its value, the lower the stability within the landscape unit and the greater the heterogeneity and discontinuity among patches [41]; *n_i_* denotes the number of patches of landscape type *i* and *A_i_* denotes the total area of landscape type *i*.
Landscape dominance index (*DO_i_*)	$DO_{i}=\frac{\left(Q_{i}+M_{i}\right)+2L_{i}}{4}$	The higher the value, the greater the influence of the landscape type on the overall landscape pattern [42]. *Q_i_* = number of samples in which patch *i* occurs/total number of samples; *M_i_* = number of patch *i*/total number of patches; *L_i_* = area of patch *i*/total area of samples.
landscape separateness index (*S_i_*)	$S_{i}=D_{i}\times \frac{A}{A_{i}}$ $,D_{i}=\frac{1}{2}\sqrt{\frac{n_{i}}{A}}$	The greater the degree of separation between different patches in a landscape type, the more discrete the distribution of the landscape type in the region for a correspondingly higher degree of fragmentation [40]; *A* is the total area of the landscape; *D_i_* is the distance index of landscape type *i*.
Landscape disturbance index (*E_i_*)	$E_{i}=aC_{i}+bS_{i}+cDO_{i}$	*E_i_* describes the extent to which ecosystems located in different landscape types are disturbed by human activities and characterizes differences related to maintenance of ecological stability of different landscape types [43]; *a*, *b,* and *c* represent weights of the corresponding landscape indices; according to results of previous studies, values of *a* = 0.5, *b* = 0.3, and *c* = 0.2 are assigned.
Landscape vulnerability index (*F_i_*)	Based on the previous studies	The higher the value, the more vulnerable and unstable the landscape type is and the more likely it will suffer ecological losses and physical changes due to external disturbances [44]. Based on the previous studies, in this study [15], vulnerability indices of six landscape types were assigned as follows: unused land 6, water 5, cultivated land 4, grassland 3, woodland 2, and residential land 1, with the landscape vulnerability index *F_i_* obtained after normalization.
Landscape loss degree index (*R_i_*)	$R_{i}=E_{i}\times F_{i}$	Ri indicates the degree of loss of natural properties of ecosystems represented by different landscape types when they are subjected to natural and anthropogenic disturbances [45].

**Table 2 ijerph-19-01595-t002:** Area and changes in land-use types in Guangxi from 2000 to 2018.

Land-Use Types	Area/km^2^			Degree of Change	
	2000	2010	2018	2000–2010	2010–2018
Arable land	51,770.71	51,516.04	50,764.22	−0.05	−0.15
Woodland	155,492.06	155,604.46	154,957.78	0.01	−0.04
Grassland	20,934.12	20,578.23	20,616.04	−0.17	0.02
Waters	3610.52	3821.93	3824.83	0.59	0.01
Construction Land	4483.9	4828.2	6136.69	0.77	2.71
Unused land	35.75	34.97	86.74	−0.22	14.80

**Table 3 ijerph-19-01595-t003:** Matrix of land-use type transfer in Guangxi from 2000 to 2010 (unit: km^2^).

2000	2010						
	Arable Land	Woodland	Grassland	Waters	Construction Land	Unused Land	Sum
Arable land	50,711.81	516.82	70.91	118.56	354.11	0.92	51,773.14
Woodland	511.40	154,601.26	183.06	104.16	123.67	0.25	155,523.80
Grassland	88.48	459.02	20,289.70	41.73	44.43	0.10	20,923.46
Waters	68.26	35.41	12.58	3484.48	8.85	0.07	3609.66
Construction Land	137.91	21.64	10.72	16.95	4270.10	0.26	4457.57
Unused land	0.18	0.37	0.42	1.37	0.02	33.27	35.64
Sum	51,518.05	155,634.52	20,567.40	3767.26	4801.18	34.86	236,323.27

**Table 4 ijerph-19-01595-t004:** Matrix of land-use types transfer in Guangxi from 2010 to 2018 (unit: km^2^).

2010	2018						
	Arable Land	Woodland	Grassland	Waters	Construction Land	Unused Land	Sum
Arable land	49,397.66	982.07	163.15	90.80	881.95	1.26	51,516.91
Woodland	1000.42	153,472.65	574.93	115.54	454.72	3.86	155,622.11
Grassland	157.92	411.25	19,834.53	34.03	127.61	0.30	20,565.64
Waters	59.04	74.80	20.80	3569.00	44.37	48.16	3816.17
Construction Land	150.15	34.06	7.86	10.37	4598.79	0.21	4801.45
Unused land	0.35	0.57	0.15	0.71	0.99	32.02	34.79
Sum	50,765.54	154,975.41	20,601.42	3820.44	6108.44	85.82	236,357.08

**Table 5 ijerph-19-01595-t005:** Landscape pattern indices of different land-use types.

Type	Year	NP	CA/km^2^	*C_i_*	*DO_i_*	*S_i_*	*E_i_*	*F_i_*
Arable land	2000	50,362	5177,071.17	0.0097	0.2541	0.1054	0.0873	0.1905
	2010	49,845	5151,604.14	0.0097	0.2522	0.1054	0.0869	0.1905
	2018	50,419	5076,421.92	0.0099	0.2490	0.1075	0.0870	0.1905
Woodland	2000	25,731	15,549,206.13	0.0017	0.4293	0.0251	0.0942	0.0952
	2010	26,040	15,560,446.23	0.0017	0.4297	0.0252	0.0943	0.0952
	2018	26,455	15,495,778.26	0.0017	0.4272	0.0255	0.0940	0.0952
Grassland	2000	24,201	2093,412.15	0.0116	0.1405	0.1806	0.0881	0.1429
	2010	24,599	2057,823.27	0.0120	0.1402	0.1853	0.0896	0.1429
	2018	25,014	2061,604.17	0.0121	0.1392	0.1865	0.0899	0.1429
Waters	2000	6615	361,052.37	0.0183	0.0622	0.5475	0.1859	0.2381
	2010	6534	382,193.28	0.0171	0.0631	0.5141	0.1754	0.2381
	2018	6364	382,483.26	0.0166	0.0629	0.5070	0.1730	0.2381
Construction Land	2000	29,817	448,390.26	0.0665	0.1100	0.9361	0.3361	0.0476
	2010	29,752	482,820.3	0.0616	0.1111	0.8685	0.3136	0.0476
	2018	30,923	613,669.05	0.0504	0.1159	0.6966	0.2574	0.0476
Unused land	2000	128	3575.43	0.0358	0.0038	7.6914	2.3261	0.2857
	2010	124	3496.68	0.0355	0.0037	7.7417	2.3410	0.2857
	2018	284	8673.75	0.0327	0.0058	4.7232	1.4345	0.2857

**Table 6 ijerph-19-01595-t006:** Guangxi ecological risk control area table.

Type of Control Area	Risk Level	Distribution Area
Key control areas	High-risk area, medium–high-risk area	Liuzhou, Laibin, Guigang, Nanning, Qinzhou, Beihai, Eastern Fangchenggang
Strict control area	Medium-risk area, low–medium-risk area	Hechi East, Guilin, Hezhou North, Wuzhou West, Yulin, Chongzuo, Baise South, Fangchenggang West
General control area	Low-risk area	Western Hechi, Southern Hezhou, Eastern Hezhou, Western Hezhou, Northern Baise

## Data Availability

Not applicable.

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
