# Peer review of "Landscape Pattern and Ecological Risk Assessment in Guangxi Based on Land Use Change"

_ijerph, 2022, doi:10.3390/ijerph19031595_

Round 1

Reviewer 1 Report

The entitled manuscript "Landscape pattern and ecological risk assessment in Guangxi based on land use change" is an interesting study on the application of landscape change indices and their relationship with potential risk estimation. 

The text is very well written and structured, very interesting to read and is sure to be a work of international impact. However, there are several points that are not entirely clear to me and which I think need to be expanded upon by the authors.

The methodology section does not explain how the different scales of information fit together to bring all the data at the same level?

In methods section this paragraph " According to the experimental results, the ecological risk value of Guangxi is between 9.74 and 12.80 and the ecological risk of the landscape in the study area can be divided into five levels of low, low-medium, medium, medium-high and high risk as visualized using ArcGIS, with corresponding assigned value intervals as follows: low risk (<10.22), low-medium risk (10.22~10.57), medium risk (10.57~10.98), medium-high risk (10.98~11.16) and high risk (>11.16)." express some results, maybe this should be placed in the Results section.

In Results section both matrix in Table 3 and 4 are unclear. A double-entry matrix should have only half of the data, in this sense the changes of land would be seen from 2000 to 2010 in Tab 3. and from 2010 to 2018 in Tab 4.  From the way the data is presented it is not clear from which type of use is changed to another type of use, bearing in mind of course that there are uses that are not reverted, if this were to happen or had been identified, perhaps the approach of the work would be different. Please this point should be explained in depth.

In terms of results, what has happened in 2018 to reduce medium-high and high risks overall? Have the lower risks been estimated to be the result of change of land use rather than environmental conditions? 

In addition, Perhaps the autocorrelation analysis should precede the results of the risk analysis, it would be good to decide a priori whether we are analysing the best variables to draw conclusions. 

Other comments: 

Please, be aware with the repetition of the key words of the title. This fact could reduce the visibility of the paper. 

It is difficult to place the comments without the line-numbers and the page numbers. In the third page there is a mistake in "...Nevertheless, rapid industrialization and urbanization in thar region [24]" that should be replaced with that.

Table 4. What does 2000/Km2 mean?

Figure 4. The colors on the maps I don't think are well chosen, they could be a scale from green for lower risk to deep red for high risk. Or another color scale that relates to the gradation from one category to another.

Figure 5. I would also change the color coding and make it uniform with that of figure 4.

Figure 6. This caption has poor quality, please improve it

Reviewer 2 Report

The article I received for review entitled: " Landscape pattern and ecological risk assessment in Guangxi based on land use change” examines the landuse/landcover changes, they authors raise the topic of climate change and anthropogenic development of areas and their negative impact on the their evaluation. This paper examines six land use types to evaluate the temporal characteristics of landscape change and ecological risks in Guangxi city. Valuable data, collected systematically allowed the authors to study changes over the years and compare them taking into account changes in land use and ecological risks. This work uses remote sensing data from land use monitoring in Guangxi from 2000, 2010 and 2018. The results obtained provide guidance for achieving ecologically sensitive regional land use management and reducing and controlling ecological risks in the city.

Overall, the authors use a broad body of literature to support their statements, however I have some general remarks. Most importantly the study has a very limited range and is mostly supported by Chinese authors an findings, even when it comes to defining the landscape, while the processes in the landscape have been studied for decades worldwide. I expect the authors to provide a broader context to their article.

I also find the structure of the article to be a bit blurry and sometimes there is little connection between the paragraphs.

  1. The authors introduce the examples of assessments of ecological risks assessments in a very broads sense, without explaining examples of what risks could potentially happen while later on they introduce the definition of the landscape out of the blue (not providing the most important body of literature in the matter of landscape change analysis that would be well recognized globally) to come up with a brief description of differences between ecological risk assessment and landscape ecological risk assessments. The differentiation, without examples is fuzzy and difficult to follow, what do the authors have in mind. Please clarify this paragraphs, and best support with examples.

Later on the authors present a whole paragraph of examples, which I find necessary to be rewritten as it consists of multiple examples The author 1 did that, and the author 2 did that…. Please shorten this paragraph only focusing on what new piece of information is relevant to this study. Why is the information that the authors used 2x2 grid of squares relevant here?

I also strongly encourage the authors to present the data on various methodologies to be tabularized and content analysis was performed on this data. This would allow the authors to find similarities and differences between the approaches used, they could for instance indicate at the differences between former and new approaches which would largely improve the manuscript, which overall, presents a very narrow piece of information. This would also support the selection of the methods used by the authors for the landscape analysis.

Information about the location of the study should be to a large extent moved to Materials and Methods

Please clarify the aims of the study more clearly – what is the novelty of this research? , the part about avoiding drawback should be moved to materials and methods.

The authors refer to remote sensing data while later on they refer to various landuse classes, as I understand the remotely-sensed data have already been classified in the monitoring system and the authors for the purpose of this study use those classified data not raw remotely-sensed images. Be specific, otherwise this is not sure whether the analysis of satellite-derived images was part of the study or you used “ready to use” data. Also for an international reader the specifics of the input data is k=not clear. Is the data derived from Landsat satellite? Please specify

Please provide a paragraph explaining the selection of landscape metrics applied.

Table 1 please add the names of the indicators in the table, it will make the table more clear to read.

I am surprised that the authors do not refer to a significant body of literature on landscape metrics. If you try google scholar the term “landscape metrics” returns over 30 000 articles containing this phrase as this is the basis of any landscape ecology analysis.

In the table 1 the authors mention “Expert scoring” while this term is not explained in the methodology. How was the objectivity of the scoring ensured?

I am sure the clarity of the article could be ensured by introducing a flowchart with all the steps of the procedure. You can find an example of such flowchart in this article:  https://doi.org/10.1016/j.ufug.2021.127155

Figure 3 – are the authors able to compute statistical differences for this graph?

I am also missing a critical review of the methods and results obtained. A paragraph on the limitations of the study would be necessary.

Last but not least, obviously through the whole discussion, the authors focus on changes and ecological security of Guangxi. While in order to be interesting to an international reader – the context of this study needs to be expanded.

Reviewer 3 Report

This manuscript deals with a relevant issue: Landscape pattern and ecological risk assessment.

The manuscript proposes an ecological risk assessment model according to both the landscape disturbance index and landscape fragility index  and apply it to describe the overall spatial pattern characteristics of ecological risk in Guangxi- China.

The research issue and objectives are clear and coherently stated.

The material and method are sound and coherent with the research objectives.

The results  presentation and discussions are sound and supported by rigorous statistical nalysis.

The conclusions are relevant and supported by the research results.

In the concluding remarks, it seems worth adding specipfic public policy implications and methodological contributions from the research results.

Reviewer 4 Report

Review, Manuscript ID: ijerph-1542148

“Landscape pattern and ecological risk assessment in Guangxi based on land use change”

The study aims to assess the patterns of dynamic spatial/temporal change in land use and ecological risks using land use data and analyse spatial correlations with ecological risks as well as exploring the effect of natural and socio-economic factors on ecological risks. The paper deals with an interesting and challenging topic that needs to be studied in different context in different cities / countries and regions.

The aim of the study is clear and relevant to the scope of the International Journal of Environmental Research and Public Health. The general structure of the paper is easy to follow; the different chapters are well balanced in length and contents. The paper presents an interesting method to address its main purposes. Also, the technical, structural and formal quality of the manuscript is good, and the results are thoroughly presented, and discussed with the relevant literature.

This being said, there are some minor corrections and changes that the authors need to address regarding the paper in its present form.

Page 5, Subsection 2.3.2 Ecological risk assessment model: the first sentence is not clear, please rewrite it, and possibly shorten the sentence to make it clearer.

Page 6, The last sentence of the page starting with…`According to the experimental results, the ecological risk value of Guangxi is`: Here is it possible to explain why you have used these intervals, any justification for this decision (eg underlying reasons or any references)?

Page 9, Subsection 3.1.2 Landscape Characterization: I am not sure if Landscape characterisation is the right term here. You might use landscape pattern but landscape characterisation is a quite different term...

Round 2

Reviewer 1 Report

Many thanks to the authors of the paper for their kind words and for taking my suggestions into account. The doubts I had raised in the first review have been resolved and I think that in the manuscript they have made a great effort to clarify different points in the Methods, Results and Discussion and Conclusions sections. Therefore, I believe that the work is worthy of publication in the IJERPH journal.
I would like to make a comment regarding Figure 7. Its quality is very poor compared to the rest of the figures in the paper. I recommend the authors to improve the quality of this figure.

Author Response

We are very grateful for your valuable comments and suggestions. Based on your suggestions, we have made revisions to the paper. The revised parts of the manuscript are highlighted in red. Our detailed point-by-point response to your comments is provided below.

Point 1: Many thanks to the authors of the paper for their kind words and for taking my suggestions into account. The doubts I had raised in the first review have been resolved and I think that in the manuscript they have made a great effort to clarify different points in the Methods, Results and Discussion and Conclusions sections. Therefore, I believe that the work is worthy of publication in the IJERPH journal.

Response 1: We are very grateful to your comments and constructive suggestions for the manuscript. According to your suggestions, we had amended the relevant part in manuscript, and your questions were answered below.

Point 2: I would like to make a comment regarding Figure 7. Its quality is very poor compared to the rest of the figures in the paper. I recommend the authors to improve the quality of this figure.

Response 2: We appreciate your careful review and suggestion. We also wanted to improve the quality of Figure 7, but the Figure 7 is automatically generated by GeoDa software, we cannot modify it.

Reviewer 2 Report

The authors took into account most of my comments. What I find little satisfactory is that instead of reformulating whole chapters they mostly added new bits without incorporating them in the text (eg the part on introducing the definition of a landscape which is not connected with the next sentence L47. Please rearead the text and ensure a better flow of the story, removing the redundant bits. 

Lines 66-74 are not supported with citations. Please shorten this bit. 

Line 150 the authors still use the term "land use remote sensing data, which I requested to be clarified, either landuse, or remote sensing data". If remote sensing, what kind of satellite and sensors were used? In Europe we mostly base such analyses on Corine Landcover data which are processes SPOT satellite remote sensing data, but these are not referred to as "land cover remote sensing data:". This causes confusion and is incorrect. 

L177 wording, this was not an experiment, this was an analysis that was performed. 

The paragraph added after the conclusions L616 does not match the conclusions. It is just the authors own elaboration (without reference to lliterature). It should be moved to the discussion and should refer to achieved results. Its fuzzy and it is hard to get what the authors had in mind. 

My biggest comment reagrding making the article sound to international audience was ensured by adding 2 references and one brief paragraph. Not what I had in mind and the authors should make more effort to address my comments. No change in the abstract was made to give even a hint of an internationally recognized phenomenon being investigated. 

I am suggesting minor revisions but I urge the authors to take the reviewers' comments more seriously into account. 
